# Understanding the Approximation Gap of Neural Networks

## Abstract

Neural networks have gained popularity in scientific computing in recent years. However, they often fail to achieve the same level of accuracy as classical methods, even on the simplest problems. As this appears to contradict the universal approximation theorem, we seek to understand neural network approximation from a different perspective: their approximation capability can be explained by the non-compactness of their image sets, which, in turn, influences the existence of a global minimum, especially when the target function is discontinuous. Furthermore, we demonstrate that in the presence of machine precision, the minimum achievable error of neural networks depends on the grid size, even when the theoretical infimum is zero. Finally, we draw on the classification theory and discuss the roles of width and depth in classifying labeled data points, explaining why neural networks also fail to approximate smooth target functions with complex level sets, and increasing the depth alone is not enough to solve it. Numerical experiments are presented in support of our theoretical claims.

## 1 Introduction

Neural networks have exhibited strong performance in many fields, including scientific computing (Sirignano & Spiliopoulos, 2018; Raissi et al., 2019) which is traditionally handled by classical methods such as finite element method (FEM) (Ern & Guermond, 2004). As it attracts more and more attention recently, a natural question arises: Do learning-based methods really outperform classical methods in practice? It has been observed that when solving one-dimensional Burgers equations, classical FEM can achieve the $L^2$ error on the magnitude of $10^{-7}$ (Khater et al., 2008), in contrast to the error of magnitude $10^{-2}$ attained by neural networks (Krishnapriyan et al., 2021; Lu et al., 2022). This gap casts doubt on the universal approximation theorem of neural networks: a two-layer neural network with sigmoid activation can approximate *any* continuous functions if it is wide enough (Funahashi, 1989; Hornik et al., 1989; Barron, 1993).

Actually, the universal approximation theorem alone may not be sufficient to explain the power of neural networks. Since the space of square-integrable functions over $D$, say $L^2(D)$, is separable (Folland, 1999), every basis can perfectly represent any function in $L^2(D)$ if it is allowed to have infinitely many parameters. Thus, the assumption of arbitrary widths is not sufficient to distinguish neural networks from existing function approximators. Inspired by this fact, many attempts have been made to understand the approximation power of neural networks by evaluating their performance under a fixed number of parameters (Berner et al., 2021; DeVore et al., 2021; Petersen et al., 2021). In particular, the approximation power of neural networks is demonstrated in the way that they can achieve the same level of error as those by classical basis functions, such as polynomials and trigonometric functions, but with fewer parameters (Liang & Srikant, 2016; Elbrächter et al., 2019; Kim et al., 2023). The error analysis is usually done under the assumption that the target function is Lipschitz continuous. However, most of these frameworks do not apply to non-smooth and discontinuous solutions which is common in scientific computing problems.

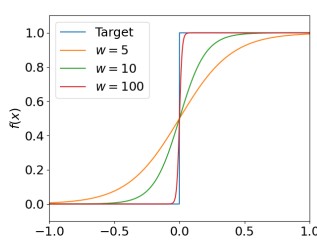

Figure 1: When approximating the step function using a *sigmoid* neuron $f(x; w) = \sigma(wx)$, the global infimum is attained at $w = \infty$.

In this paper, we show that one of the fundamental distinctions between neural networks and traditional function approximators like polynomials and Fourier series is the topology of their image sets in function space. As every bounded subset of the image set is compact for classical schemes, the intersection of the image set of a neural network and the closed unit ball is *not* compact in function space. Therefore, a Cauchy sequence in the image set $Im(f)$ may converge to some function that is not contained in $Im(f)$. This implies that having a global minimum at infinity is possible especially when the target function has discontinuity as shown in Figure 1. Furthermore, as we will show in Section 4, the global minimum of approximation may not exist when the target function is not continuous. A major issue follows immediately is that gradients may vanish or blowup (Glorot & Bengio (2010); Gallon et al. (2022)) as it approaches the optimal solution, depending on what activation function is used in practice.

Unlike many traditional methods that often have closed-form solutions, neural network approximation relies on gradient-based algorithms, which are local optimizers. Hence, focusing on the theoretical infimum might be of little practical significance if it can hardly be reached through gradient descent. One would possibly argue that even if the global minimum does not exist, it should still be able to achieve arbitrarily small errors by letting weights approach infinity. Unfortunately, it is impossible in practice: when the grid size is $\delta$, the maximum attainable weights in a neural network that can be attained through gradient descent scale as $\mathcal{O}(\delta^{-1})$ as $\delta$ approaches 0 due to the presence of machine precision. This causes a minimal attainable $L^2$ error of order $\mathcal{O}(\sqrt{\delta})$ when the global minimum does not exist. Besides, improving the machine precision cannot significantly reduce the attainable error. For instance, when using the *sigmoid* or *tanh* activation functions, the minimum achievable error decays at a rate of $\mathcal{O}(\frac{1}{\sqrt{p}})$ where $\epsilon = 2^{-p}$ gives the machine precision. These results partially explain why neural networks cannot achieve an approximation error below certain thresholds in solving certain class of partial differential equations (PDEs) that have discontinuous solutions.

Another gap between the approximation theory and reality is that while deep networks can provably approximate periodic functions in a more efficient way than shallow networks (Elbrächter et al., 2019), it is often observed that they still fail to achieve good performance in practice (Schultz et al., 2021; Chen et al., 2022). Notice that approximating a function is equivalent to classifying its level sets, we aim to understand this gap from the perspective of classification theory (Minsky & Papert, 1969; Bishop, 1995; Koiran & Sontag, 1997; Sontag, 1998; Montúfar et al., 2014; Bartlett et al., 2019). Specifically, there are two steps in classification: first, separating data points with different labels; second, gluing points with same label together. We show that the separating task is handled by the width, and the gluing part is done by the depth. This result suggests that the target function whose level sets are highly disconnected are naturally hard to approximate and increasing the depth alone is not sufficient to fit them well.

Our main contribution in this paper can be summarized as follows:

- We investigate the approximation gap of neural networks in scientific computing, especially for approximating non-smooth and discontinuous solutions in PDEs.

- We show that for a neural network, a bounded subset of its image set does not necessarily have compact closure, especially in the case of discontinuous target functions, which further implies the non-existence of the global minimum;

- Due to the machine precision, the weights cannot grow arbitrarily large, which then explains the approximation gap of neural networks.

- We discussed that increasing depth alone is insufficient to improve the approximation accuracy when the level sets of the target function is complex.

Further discussion on related topics can be found in the appendix, along with some open problems that may be of interest for future study.

## 2 RELATED WORK

**Loss landscape of neural networks.** Due to the lack of a closed-form solution, the global minimum of neural network-based PDE solvers cannot be calculated directly. Thus, gradient-based methods lie at the core of optimizing these problems, which further motivates the study of loss landscapes.

By treating neural networks as a mapping from parameter spaces to function spaces, some works investigate the topological properties of the parameterization mapping (Levin et al., 2022; Nouiehed & Razaviyayn, 2022). Others study the topology of the landscape, including properties of saddle points (Kawaguchi, 2016), geometry of gradient flow (Zhao et al., 2023), existence of spurious minima (Venturi et al., 2019; Pollaci, 2023; Nguyen et al., 2019), connectedness of sublevel sets (Freeman & Bruna, 2017), and the dimension of minima in overparametrized networks (Cooper, 2021). Characterizing the loss landscape of neural networks is crucial to understanding the capacity and limitation of deep learning. We contribute to this line of work by providing insights on the topology of image sets. In particular, when the global minimum does not exist, all local minima are spurious, a scenario that has received very little attention.

**Limitation of ML methods in solving PDEs.** Solving partial differential equations (PDEs) is one of the core areas in scientific computing. A number of deep learning algorithms has been developed to learn PDE solutions, such as the deep Galerkin method (Sirignano & Spiliopoulos, 2018), physics-informed neural networks (PINN) (Raissi et al., 2019), Fourier neural operator (Kovachki et al., 2021b), and DeepONet (Lu et al., 2019; 2022). Although some of these methods are proven to have the universal approximation property (Lu et al., 2019; Kovachki et al., 2021a), they are often not compared to traditional methods such as FEM, whose error provably approaches zero as the resolution increases. Additionally, PINN has worse accuracy compared to traditional computational fluid dynamics methods (Cai et al., 2021) and can fail to produce a physical solution in certain problems (Chuang & Barba, 2022). This limitation may be attributed to spectral bias and different convergence rates of different loss components (Wang et al., 2022), and even PINN with enough expressiveness can fail to optimize (Krishnapriyan et al., 2021). In this paper, we discuss another possible limitation caused by the absence of a global minimum.

## 3 NEURAL NETWORK APPROXIMATION

Generally, given a representation $f : \mathbb{R}^N \to L^2(D)$ where $D \subset \mathbb{R}^d$ is a compact domain, finding the best approximation for a target function $g \in L^2(D)$ is equivalent to seeking the solution of the corresponding optimization problem

$$\theta^* \in \operatorname*{arg\,min}_{\theta \in \mathbb{R}^N} \|f(\cdot; \theta) - g\|_2.$$

Let us $J(\theta)$ denote the $L^2$ error $\|f(\cdot; \theta) - g\|_2$ thereafter. To study the approximation of discontinuous functions, we start with a special basis in $L^2(D)$: the set of all indicator functions of hypercube. More precisely, every function in $L^2(D)$ can be approximated arbitrarily well by a finite linear combination of those functions:

**Proposition 3.1.** *(Folland, 1999) For any $g \in L^2(D)$ and $\epsilon > 0$, there exists $h = \sum_{i=1}^{K} a_i \chi_{R_i}$ where $K \in \mathbb{N}$ and each $R_j$ is a product of intervals, such that $\|g - h\|_2 < \epsilon$.*

Consider the following set

$$S_K(d) = \{g = \sum_{i=1}^{K} a_i \chi_{R_i} : a_i \in \mathbb{R}, R_j \text{ is a product of intervals in } \mathbb{R}^d\}. \tag{1}$$

According to Proposition 3.1, the union $\cup_{i=1}^{\infty}(L^2(D) \cap S_K(N))$ is dense in $L^2(D)$. Therefore, by relating neural networks to the functions in $S_K(d)$ for every integer $K$, we can demonstrate the approximation power of neural networks:

**Theorem 3.1.** *Suppose that the neural network $f$ has at least 2 hidden layers and let $L$ denote its width. For any $K \in \mathbb{N}$, the closure $\overline{Im(f)} \supset S_K(d)$ if*

- *the activation function $\sigma(\cdot)$ is sigmoid and $L \geq 2Kd$;*

- *the activation function $\sigma(\cdot)$ is Relu and $L \geq 4Kd$.*

*Proof.* This proof consists of two parts: first, we prove the claim for *sigmoid* networks; second, we will show how the case of *Relu* networks can be reduced to the first case.

*Sigmoid:* Let $\sigma$ be *sigmoid*. For any hypercube $R = [l_1, r_1] \times ... \times [l_d, r_d]$, consider the network

$$f(x; \theta_n) = \sigma(ny - n(d + \frac{1}{2})), \quad y = \sum_{i=1}^{d}(\sigma(w_{i,1}^T x + b_{i,1}) + \sigma(w_{i,2}^T x + b_{i,2}))$$

where $w_{i,1} = [0, ..., 0, n, 0, ..., 0]^T$ and $w_{i,2} = [0, ..., 0, -n, 0, ..., 0]^T$ are aligned with the $i$-th axis, $b_{i,1} = -nl_i$ and $b_{i,2} = nr_i$. It can be verified that the limit of the sequence $\{f(x; \theta_n)\}_{n=1}^{\infty}$ exists for almost every $x \in \mathbb{R}^d$, which is exactly the indicator function $\chi_D$. Also, since $f(x; \theta_n) \in Im(f)$ for all $n$, we have $\chi_D \in \overline{Im(f)}$. This can be easily extended to the case of $K$ sums.

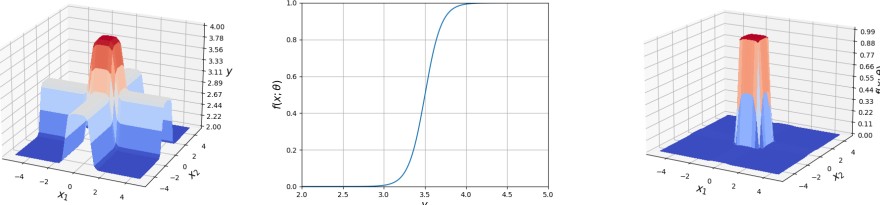

(a) The output of the first hidden layer.  (b) The output of the second hidden layer.  (c) The output of the entire neural network.

Figure 2: Illustrations of how $f(x; \theta_n)$ converges to $\chi_D(x)$, where $D = [-1, 1] \times [-1, 1]$ and $n = 10$.

*Relu:* Without loss of generality, it suffices to show that the indicator function of interval $[0, 1]$ is approximated by a sequence of *Relu* networks $\{f(x; \theta_n)\}$ that are given by

$$f(x; \theta_n) = \sigma(y - 2 + \frac{1}{n}), \quad y = \sigma(nx + \frac{1}{2}) - \sigma(nx - \frac{1}{2}) + \sigma(-nx + n - \frac{1}{2})$$

so that $f(x; \theta_n) \to \chi_{[0,1]}$ a.e. $x \in [-1, 1]$ as $n \to \infty$. This can be extended toward higher-dimensional cases, and thus we omit the detail. $\qquad \square$

From Theorem 3.1, the following universal approximation theorem for neural networks is immediate:

**Corollary 3.1.** *Suppose that the neural network $f$ has at least 2 hidden layers with sigmoid or Relu activation function. Let $L$ denote its width, then for any $h \in L^2(D)$ and $\epsilon > 0$, there exists $L_h \in \mathbb{N}$ such that $\inf_{\theta \in \mathbb{R}^N} \|f(\cdot; \theta) - h\|_2 < \epsilon$ when $L \geq L_h$.*

Note that this corollary does not imply that neural networks can always approximate functions arbitrarily well. As we have seen in Figure 1, the global minimum of approximating step functions is at infinity, and hence does not exist. Thus, the actual error that can be achieved in practice may not be as small as suggested in the previous corollary. We will discuss more details in the next part.

## 4 UNDERSTANDING THE APPROXIMATION GAP

In this section, we demonstrate that the closure of a bounded subset of $Im(f)$ is not necessarily compact in $L^2(D)$, which poses a serious challenge to the existence of global minimum. We then derive a lower bound for the attainable error that characterizes the approximation gap.

### 4.1 THE TOPOLOGY OF IMAGE SET $Im(f)$

Although every closed and bounded set in finite-dimensional Euclidean space is compact, it is no longer true when the underlying space is infinite-dimensional. Therefore, compactness is no longer a trivial implication of boundedness plus closedness. Since unbounded sets are never compact, we will focus on its intersection with the closed unit ball $\mathcal{B}$, say $\overline{Im(f)} \cap \mathcal{B}$, to better characterize the compactness of the closure of image set $\overline{Im(f)}$. First, we introduce the following concept:

**Definition 4.1.** *(Radial unboundedness) A mapping $g : \mathbb{R}^N \to L^2(D)$ is called radially unbounded if for any $M > 0$, there exists $R > 0$ such that $\|g(\theta)\|_2 \geq M$ for all $|\theta| \geq R$.*

All subspace approximations of the form $g = \sum_{i=1}^{K} a_i g_i$ are radial unbounded when the basis functions $g_i$ are linearly independent. The next theorem shows that its image set must be "simple" in this case:

**Theorem 4.1.** *Suppose that $g : \mathbb{R}^N \to L^2(D)$ is continuous and radially unbounded, then $\overline{Im(g)} \cap \mathcal{B}$ is compact.*

The theorem implies that for any target function $\phi \in L^2(D)$, the global minimum of $\min_\theta \|g(\theta) - \phi\|_2$ always exists in this case. Although not every compact set in $L^2(D)$ can be embedded into a finite-dimensional space, the following result says that they are "almost" finite-dimensional in the sense that for any given compact $S \subset \mathcal{B}$, there always exist functions that are "far away" from set S:

**Theorem 4.2.** *Let $S \subset \mathcal{B}$ be a compact set, then for any $\epsilon \in (0, 1)$, there exists $g \in \mathcal{B}$ such that $\inf_{h \in S} \|g - h\|_1 \geq 1 - \epsilon$.*

To get a better sense of how a compact set in $L^2(D)$ looks like, the following theorem provides a full characterization of when a bounded set has compact closure in function space:

**Proposition 4.1.** *(The Fréchet-Kolmogorov Theorem, Brezis (2010)) Let $1 \leq p < \infty$ and $D \subset \mathbb{R}^d$ be a bounded measurable set. Then, for any bounded set $S \subset L^p(D)$, its closure $\bar{S}$ is compact if and only if for any $\epsilon > 0$, there exists $\delta > 0$ such that $\|f(x + h) - f(x)\|_p < \epsilon$ for all $f \in S$ and all $h \in \mathbb{R}^d$ with $|h| < \delta$.*

The above condition is equivalent to the statement that the functions in $S$ have $\lim_{|h| \to 0} \|f(x + h) - f(x)\|_p = 0$ uniformly, indicating that $S$ cannot contain both smooth and discontinuous functions. Now turn to consider the image set of a neural network whose activation function is either *sigmoid* or *Relu*. Since Theorem 3.1 states that the closure of its image set contains step functions, the non-compactness of $\overline{Im(f)} \cap \mathcal{B}$ follows immediately if the set of such step functions is non-compact:

**Lemma 4.1.** *The set $S_K(d) \cap \mathcal{B}$ is not compact in $\mathcal{B}$ for all $K, d \in \mathbb{N}$.*

This leads us to the following result that fundamentally distinguishes neural networks from most classical approximation schemes:

**Corollary 4.1.** *Let $f : \mathbb{R}^N \to L^2(D)$ be a neural network with sigmoid or Relu activation function, having a width greater than or equal to 2 and at least 2 hidden layers. Then, $\overline{Im(f)} \cap \mathcal{B}$ is not compact.*

Finally, there is a serious consequence: a Cauchy sequence $\{f(x; \theta_n)\} \subset Im(f)$ is no longer guaranteed to converge toward any point in $Im(f)$ due to its non-compactness, which implies that it is likely that some loss function attains its infimum at the boundary so that no global minimum exists.

**Conflict with regularization terms.** Regularization is a standard practice in machine learning, as it improves generalization and stabilizes training. However, after introducing a regularization term, we will never be able to reach the global minimum if it is at infinity. For instance, consider the $L^2$-regularized loss function

$$L(\theta) = J(\theta) + \epsilon|\theta|_2^2$$

where $\epsilon > 0$. Suppose $J(\theta)$ is lower bounded by some constant $C$, then we have $L(\theta) \to \infty$ as $|\theta| \to \infty$. This implies that the global minimum of $L(\theta)$ must exist with a finite norm. In other words, the solution of the original optimization problem is no longer the solution of the regularized problem. It is very likely to happen when solving certain class of PDEs that have only discontinuous solutions (Evans, 2010).

## 4.2 The Unreachable Infinity

Theoretically, even if the global minimum is at infinity, we can still get arbitrarily small error using sufficiently large weights. However, as the weights grow, the effective regions of corresponding neurons shrink until they cover no data points. That poses an upper limit to the norm of weights, which then yields the minimum of attainable error.

Consider a finite dataset $X = \{x_1, ..., x_N\} \subset \mathbb{R}^d$, and a neuron $f(x; \theta) = \sigma(w^T x + b)$ where $\sigma$ is the *sigmoid* activation function. Since the precision of a computing machine is always finite, let it be $2^{-p}$ where $p \in \mathbb{N}$. Then for any quantity $\epsilon \in \mathbb{R}$ with $|\epsilon| < 2^{-p}$, it yields $\epsilon = 0$ on the machine.

Let $\phi(\cdot)$ be the target function with $\max_{x \in D} |\phi(x)| \leq M$. For each $x \in D$, the norm of the gradient of approximation error $L(x, \theta) = (f(x; \theta) - \phi(x))^2$ with respect to $w$ is

$$|\nabla_w L(x, \theta)| = 2|f(x; \theta) - \phi(x)| \, |\frac{2xe^{-(w^T x + b)}}{(1 + e^{-(w^T x + b)})^2}| \leq 2(M + 1)M' e^{-|w^T x + b|}$$

when $M' = \max_{x \in D} \|x\|$. Similarly, we have $|\nabla_b L(x, \theta)| \leq 2(M + 1)e^{-|w^T x + b|}$.

Therefore, $\nabla_\theta L(x, \theta)$ is zero on the machine when both quantities are less than $2^{-p}$ i.e., there exists

$$\delta = \max\{(p - 1)\log 2 - \log((M + 1)M'), (p - 1)\log 2 - \log(M + 1), 0\}$$

such that $\nabla_\theta L(x_i, \theta) = 0$ for all data point $x_i$ outside the region $\mathcal{E}(w, b) = \{x \in \mathbb{R}^d : |w^T x + b| \geq \delta\}$. In other words, such points have no contribution to the gradient of the loss objective with respect to the parameter of this neuron. Notice that $\mathcal{E}(w, b)$ is exactly the collection of all points whose distance to the hyperplane $w^T x + b = 0$ is $\frac{\delta}{|w|}$ where the constant $\delta$ depends only on the precision and the target function, the width of effective region $\mathcal{E}(w, b)$ decays as $\|w\|$ grows at the rate of $\mathcal{O}(\|w\|^{-1})$.

Suppose the sample domain $D$ is partitioned into equal grids of length $\delta > 0$, the gradient of the MSE $J(\theta)$ is actually estimated by

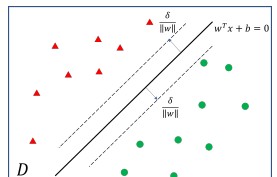

$$\nabla J(\theta) \simeq \frac{1}{N} \sum_{i=1}^{N} \frac{\partial L}{\partial \theta}(x_i, \theta)$$

which means that the parameters will stop updating when the effective region is too small to contain any data points. For instance, consider the approximation of the target function $\phi = \chi_{[0,1]}$ over the domain $D = [-1, 1]$ using a single neuron $f(x; w) = \sigma(wx)$ as in Figure 1, so that the global minimum is at $w = \infty$ with zero loss. Now consider the partition $x_i = -1 + \frac{i-1}{K}$ for $i = 1, 2, ..., 2K + 1$, according to the previous results, the largest reachable value of $\|w\|$ by gradient descent cannot exceed $\frac{(p-1)\log 2}{\Delta x}$ where $\Delta x = \frac{1}{K}$ is the grid size.

Figure 3: The effective region $\mathcal{E}(w, b)$.

On the other hand, the actual $L^2$ error at $w' = \frac{(p-1)\log 2}{\Delta x}$ is

$$\|f(\cdot; w') - \phi\|_2 = \sqrt{2}\sqrt{\frac{1}{w'}(\log(\frac{2}{1 + e^{-w'}}) + \frac{1}{2} - \frac{1}{1 + e^{w'}})} \simeq \sqrt{\frac{1 + 2\log 2}{(p - 1)\log 2}}\sqrt{\Delta x} \quad (2)$$

when $|w'| \gg 1$. It indicates that the error of approximating discontinuous functions using neural networks is of order $\mathcal{O}(\sqrt{\Delta x})$. This result is summarized as follows:

**Remark 4.1.** *Let $f$ and $\phi$ be a neural network with saturated activation function and the target function, respectively. Suppose that the target function $\phi$ has discontinuity, then the smallest attainable $L^2$ error $\epsilon \sim \mathcal{O}(\sqrt{\Delta x})$ where $\Delta x \ll 1$ is the grid size.*

Therefore, this result may suggest that the discretization error is a fundamental limitation that cannot be simply overcome by the use of neural networks against classical methods (Aubin, 2000).

## 5 FROM THE PERSPECTIVE OF CLASSIFICATION

This section focuses on another approximating gap: increasing depth alone is insufficient to achieve high accuracy. We will begin with perceptrons and explore the roles of width and depth in this case and then extend these results to ReLU networks.

### 5.1 CLASSIFYING THE LEVEL SETS

A multi-layer perceptron can be viewed as a special structure of neural networks with the binary activation function

$$\sigma(x) = \begin{cases} 1, & x \geq 0; \\ 0, & x < 0. \end{cases}$$

According to Proposition 3.1, a network $h$ achieves $\|g - h\|_2 < \epsilon$ if one can find a classifier $h$ such that

$$h(x) = \begin{cases} a_i, & x \in R_i; \\ 0, & \text{otherwise}, \end{cases}$$

which establishes the equivalence between classification and approximation. Now consider a perceptron

$$y = W_{K+1}\sigma(W_K\sigma(...W_2\sigma(W_1x + b_1) + b_2)... + b_K)$$

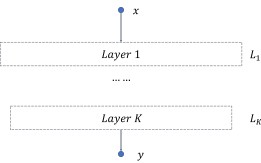

with $K$ hidden layers. Let $L_i$ and $y_i \in \{0,1\}^{L_i}$ denote the width and output of its $i$-th layer, respectively. In particular, the first layer maps the relative position of input $x \in \mathbb{R}^d$ with respect to the neurons in layer 1 to a $L_1$-dimensional binary vector $y_1$. For instance, the relative position of $x$ in Figure 5 is $y_1 = [1, 1, 0]^T$ where the arrows point towards the positive side of each neuron. Similarly, the $i$-th layer maps the $L_i$-dimensional binary vector $y_i$ to the $(i + 1)$-th layer $y_{i+1}$ based on its relative position with respect to the neurons in layer $i + 1$.

Figure 4: A K-hidden-layer perceptron.

Therefore, a classification task consists of two parts: separating and gluing. For example, suppose that we are classifying the region $D = \cup_{i=1}^4 D_i$ in Figure 6 using a perceptron. The first layer separates the entire plane into 7 regions which are mapped to the vertices on a cube. The second layer glues the vertices of $D_2, D_3, D_4$ together, so that we can find a straight line to precisely separate $D$ from other regions. Intuitively, adding more neurons to a layer enhances the perceptron's ability to separate different vertices, while adding an additional layer can glue vertices of same label together. To make it rigorous, we define the isolable sets motivated from the concept of shattering (Shai & Shai, 2014):

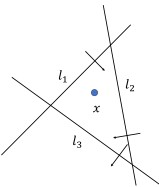

Figure 5: Represent the position of $x$.

**Definition 5.1.** *(Isolable sets) Let $X = \{x_1, ..., x_N\} \subset \mathbb{R}^d$ be a finite data set and $\mathcal{S}$ be a class of sets in $\mathbb{R}^d$. We say that $X$ is isolable by $\mathcal{S}$ if for every $x_i \subset X$, there exists $A_i \in \mathcal{S}$ such that $\{x_i\} = A_i \cap X$. In particular, $X$ is called isolable by a measurable mapping $g : \mathbb{R}^d \to \mathbb{R}^k$ if $X$ is isolable by the set $\Sigma(g) = \{g^{-1}(B) \subset \mathbb{R}^d : B \text{ is a Borel set of } \mathbb{R}^k\}$.*

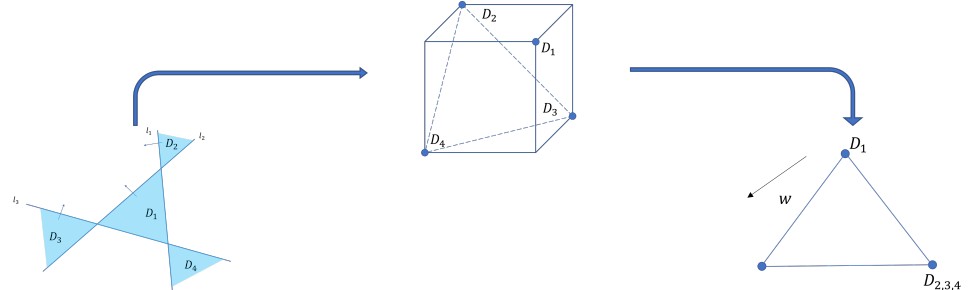

Figure 6: Illustration of how a perceptron separates the entire plane and glues all different parts of $D$ together through a composition of layers, so that data points with different labels are linear separable.

The next theorem states that adding more layers does not enhance a perceptron's isolation capability:

**Theorem 5.1.** *Let $\Phi_1 : \mathbb{R}^d \to \mathbb{R}^k$ and $\Phi_2 : \mathbb{R}^k \to \mathbb{R}^m$ be measurable mappings, and $X = \{x_1, ..., x_N\}$ be a finite set. Suppose that $X$ is not isolable by $\Phi_1$, then $X$ is not isolable by $\Phi_2 \circ \Phi_1$.*

The proof is immediate from the fact that $\Sigma(\Phi_2 \circ \Phi_1) \subset \Sigma(\Phi_1)$. Since a neural network can be regarded as a composition of layers, the following corollary is obvious:

**Corollary 5.1.** *Let $X = \{x_1, ..., x_N\} \subset \mathbb{R}^d$ be a finite set and $L \in \mathbb{N}$. Suppose that for any $f(x) = \sigma(W_1x + b_1)$ where $W_L \in \mathcal{M}(L \times d)(\mathbb{R})$ and $b_1 \in \mathbb{R}^L$, the set $X$ is not isolable by $f$. Then, for any perceptron $f$, $X$ is not isolable by $f$ if the width of its first hidden layer $L_1 \leq L$.*

It may seem counterintuitive; however, Theorem 5.1 claims that the isolation ability of a perceptron is bounded by the width of its first hidden layer. Therefore, when approximating a function with complex level sets, the width of neural network should be large, otherwise it cannot characterize their structure. It explains why neural networks have difficulty on approximating fluctuating functions.

## 5.2 CONNECTION TO COMPUTATIONAL LEARNING THEORY

The computational complexity of neural networks has been studied in the field of computational learning theory. Therefore, it is beneficial to compare the previous results with existing frameworks. First, the VC dimension is given as follows:

**Definition 5.2.** *(VC dimension, shattering) Let $X = \{x_1, ..., x_N\} \subset \mathbb{R}^d$ be a finite data set and $\mathcal{S}$ be a class of sets in $\mathbb{R}^d$. We say that $X$ is shattered by $\mathcal{S}$ if for any subset $X' \subset X$, there exists $A \in \mathcal{S}$ such that $X' = A \cap X$. Then, the VC dimension of $\mathcal{S}$ is the cardinality of the largest set that can be shattered by $H$. If the cardinality can be arbitrarily large, the VC dimension is $\infty$.*

Generally, VC dimension is an important measure of a function's capacity to classify binary labeled data. It is proved that for a network with *Relu* activation function, the VC dimension is $\mathcal{O}(WL\log(W))$ where $W$ is the number of weights and $L$ is the number of layers (Bartlett et al., 2019). At first glance, it may appear contradictory to the claim that the separation ability of a network is determined by the width of the first layer, regardless of the total number of layers. The underlying reason lies in the difference between the concept of shattering and separability. While VC dimension characterizes the largest possible set that can be shattered by the class $\mathcal{S}$, isolability determines whether a given set can be partitioned into single-point sets. Obviously, the concept of isolability is strictly weaker than shattering, the next lemma follows immediately:

**Lemma 5.1.** *Let $\Phi_1 : \mathbb{R}^d \to \mathbb{R}^k$ and $\Phi_2 : \mathbb{R}^k \to \mathbb{R}^m$ be measurable mappings, and $X = \{x_1, ..., x_N\}$ be a finite set. Suppose that $X$ is not isolable by $\Phi_1$, then $X$ is not shattered by $\Phi_2 \circ \Phi_1$.*

If the width of the first layer $L_1$ is not large enough to isolate a finite set $X$, the neural network $f$ cannot shatter $X$, regardless of the number of layers connected to it. On the other hand, it has been proved that when the width of the second layer is $2^{L_1}$, this two-hidden-layer neural network can shatter any sets that can be isolated by the first layer (Gibson & Cowan, 1990). Combining it with Lemma 5.1 yields the following result:

**Theorem 5.2.** *Let $L \in \mathbb{N}$ and $X \subset \mathbb{R}^d$ be a finite set. The following statements are equivalent:*

- *There exists a one-hidden-layer ReLU network, denoted as $f$, with a hidden layer width of $L$, such that $X$ can be isolated by $f$;*

- *There exists a multi-layer ReLU network, denoted as $f$, with a first hidden layer width of $L$, such that $f$ can shatter the set $X$.*

Hence, the argument that "deep networks can approximate certain functions with fewer parameters" does not fully justify the advantage of deep networks over shallow ones, as the role of width cannot be easily replaced by the depth.

## 6 EXPERIMENTS

We conduct numerical experiments to support our claims on the neural network approximation gap.

**Numerical solutions of PDEs.** We consider the Burgers' equation in one dimension (Basdevant et al., 1986):

$$\frac{\partial u}{\partial t} + \mu u \frac{\partial u}{\partial x} = \nu \frac{\partial^2 u}{\partial x^2}, \qquad \mu = -1, \nu = 10^{-3}, x \in [-1, 1], t \in [0, 1]$$

The Burgers' equation is a nonlinear partial differential equation that models a combination of advection and diffusion. The initial condition is $u(x, 0) = \sin\left(\frac{\pi x}{2}\right)$, which leads to a discontinuous solution. We compute the numerical solution using the spectral method implemented in Binder (2021). The time resolution is 0.01 and spatial resolution is 0.001. Figure 7(a) visualizes the solution. We evaluate the mean squared error of fully connected networks with *sigmoid* activation trained under

SGD. In the first set of experiment, we use a 2 layer network with hidden size ranging from 2 to 256. In the second set, we vary the number of layers while keeping all hidden dimension at 4. Figure 7 (b,c) shows the approximation error of neural networks with different width and depth, and the best achieved $L^2$ error exceeds the square root of the resolution ($\delta = 10^{-3}$), which further validates the claim of Remark 4.1.

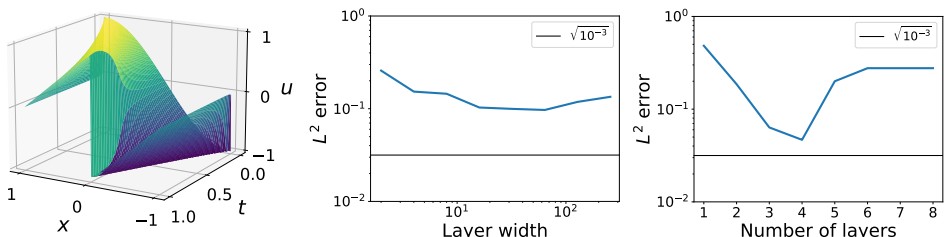

(a) Numerical solution of Burgers equation.   (b) Neural network approximation error vs width.   (c) Neural network approximation error vs number of layers.

Figure 7: Burgers equation's solution and approximation error of neural networks.

**Classfying the Smith-Volterra-Cantor set.**   Let $\mathcal{C} \subset [0, 1]$ denote the Smith-Volterra-Cantor set, which is a generalization of the famous Cantor set. It has some "strange" properties: (i) $\mathcal{C}$ is a closed set; (ii) $\mathcal{C}$ has no interior point; (iii) the Lebesgue measure of $\mathcal{C}$ is $\frac{1}{2}$. Although it requires infinite width to be perfectly represented, the $L^2$ error of fitting decays at a rate of $\mathcal{O}(L^{-1})$ where $L$ is the width of network. However, as shown in Figure 8b, the actual error attained by a two-hidden-layer *sigmoid* neural network is far higher than the theoretical infimum.

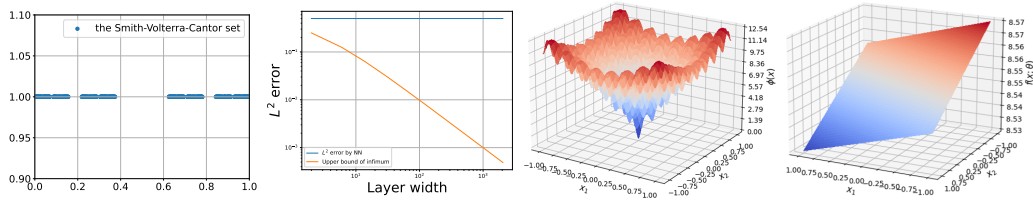

(a) The Smith-Volterra-Cantor set.   (b) Attained error and theoretical infimum.   (c) The true Ackley function.   (d) Fitting result by neural network.

Figure 8: Fitting results of different target functions.

**Approximation of the Ackley function.**   Consider a smooth target function

$$\phi(x) = -20e^{-0.2\sqrt{8(x_1^2 + x_2^2)}} - e^{0.5(\cos(8\pi x_1) + \cos(8\pi x_2))} + e + 20, \quad x_1, x_2 \in [-1, 1], \quad (3)$$

whose level sets are highly complicated and consist of many connected components as in Figure 8c. In Figure 8d, it turns out that even though the target function does not have any discontinuity, neural networks still fail to approximate due to the complexity of its level sets.

## 7    CONCLUSION

In this work, we discuss some fundamental limitations of the application of neural networks in scientific computing that hinder achieving higher accuracy. However, these issues rarely pose a concern for classical methods. Therefore, we must rethink of the use of neural networks in the field of scientific computing. Besides, although we focus on the aspect of approximation power, it's important to note that various factors, such as loss function design and optimizer choice, can also influence the performance of a numerical scheme. Overall, we believe that a comprehensive understanding on neural networks is crucial to the future of deep learning and we put some interesting open problems in the Appendix.

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

## A  NOTATION

Below is a summary of symbols used throughout this paper for reference.

- $f(x; \theta)$: The multi-layer fully-connected neural network with input $x \in \mathbb{R}^d$ and parameter $\theta \in \mathbb{R}^N$;
- $L^2(D)$: The space of square-integrable functions over compact domain $D \subset \mathbb{R}^N$;
- $C(D)$: The space of continuous functions on $D$;
- $\mathcal{B} = \{f \in L^2(D) : \|f\|_2 \leq 1\}$: The closed unit ball in $L^2(D)$;
- $\bar{S}$: The closure of $S$;
- $\mathcal{M}_{n \times m}(\mathbb{R})$: The set of all $n \times m$ real-valued matrices;
- $\|f\|_p = (\int_D |f(x)|^p \, dx)^{\frac{1}{p}}$: The $L^p$-norm of $f$ with $1 \leq p < \infty$;
- $\chi_D$: The indicator function of set $D$;
- Saturated functions: The *sigmoid* ($\sigma(x) = \frac{1}{1+e^{-x}}$) and *tanh* ($\sigma(x) = \frac{e^x - e^{-x}}{e^x + e^{-x}}$) activation functions;

## B  DEPARTING FROM CLASSICAL APPROXIMATION THEORY

In this part, we will briefly introduce how classical methods approximate functions and why those techniques are not applicable to neural networks.

**Closed-form solution.**  In general, solving the corresponding linear equations provides the closed-form solution for the optimal point, as shown in the following three examples:

**Example B.1.** *(Interpolation) Suppose $h \in C([0,1])$ is a continuous function over $[0,1]$, and $0 \leq t_0 < t_1 < ... < t_n \leq 1$ are the nodes at which we wish to interpolate $h$ by a polynomial of degree $n$, say $p(x; \theta) = \sum_{i=0}^{n} a_i x^i$ where $\theta = [a_0, ..., a_n]^T$. Therefore, it is equivalent to solving the linear system*

$$V(t_0, ..., t_n)\theta = Y$$

*where $V(t_0, ..., t_n)$ is the Vandermonde matrix and $Y = [h(t_0), ..., h(t_n)]^T$.*

**Example B.2.** *(Linear least squares) Consider a set of data $\{(x_i, y_i)\}_{i=1}^K$ and a linear regression model $g(x; \theta) = \sum_{j=1}^M a_j \phi_j(x)$ where $\theta = [a_1, ..., a_M]^T$ and $\phi_j(x)$ are basis functions. To minimize the mean-square error*

$$J(\theta) = \frac{1}{K} \sum_{j=1}^K \|g(x_i; \theta) - y_i\|^2.$$

*Since $J(\theta)$ is quadratic with respect to $\theta$, solving $\nabla J(\theta) = 0$ directly yields the best approximation*

$$\theta^* = (X^T X)^{-1} X^T Y$$

*where $X = (X_{ij})_{K \times M}$ with $X_{ij} = \phi_j(x_i)$ and $Y = (Y_i)_{K \times 1}$ with $Y = [y_1, ..., y_K]^T$.*

**Example B.3.** *(Subspace approximation) Let $h \in L^2(D)$ be the target function and we want to approximate it by $g(x; \theta) = \sum_{j=1}^M a_j \phi_j(x)$ where $\phi_j \in L^2(D)$ are linearly independent and $\theta = [a_1, ..., a_M]^T$. If we denote the subspace spanned by $\{\phi_j\}$ by $\mathcal{M}$, the best approximation $\theta^*$ is exactly the projection of $g$ onto $\mathcal{M}$. Therefore, we have $\langle h - g(\cdot; \theta^*), \phi_j \rangle = 0$ for all $j = 1, ..., M$, which is equivalent to solving the linear equations*

$$A\theta^* = b$$

*where $A = (A_{ij})_{M \times M}$ with $A_{ij} = \langle \phi_i, \phi_j \rangle$ and $b = (\langle h, \phi_j \rangle)_{M \times 1}$. In particular, we have $\theta^* = b$ if $\{\phi_i\}$ is a orthonormal basis.*

**The Gauss-Newton method.** For general nonlinear least-squares problems, however, there is no closed-form solution to work on. Therefore, gradient-based algorithms are the only way to solve them, and neural network approximation falls into this category. A common way to solve is using Gauss-Newton method when the objective is of the form $S(\theta) = \sum_{i=1}^K r_i^2(\theta)$. Let $r = [r_1, ..., r_K]^T$ denote the vectorized function, the iteration is given by

$$\theta_{n+1} = \theta_n - (J_r^T J_r)^{-1} J_r^T r(\theta)$$

where $J_r$ is the Jacobian of $r$ at $\theta = \theta_n$. In the case of convex objective function, it has been proved that the Gauss-Newton method can converge to the global minimum with a quadratic rate (Burke & Ferris, 1995). However, the Gauss-Newton method is not necessarily locally convergent, which requires the objective function to be strictly convex in a neighborhood of the local minimum (Dennis & Schnabel, 1996).

When it comes to neural networks, however, there are two general facts that prompt us to adopt an entirely different viewpoint:

- There is no closed-form solution for neural networks, or too expensive to use;
- If the global minimum is at infinity and the gradient is about to vanish, the matrix $J_r^T J_r$ can be singular and hence not invertible.

## C  THE EXPRESSIVENESS-REGULARITY TRADE-OFF

In DeVore et al. (2021), the authors conjectured that neural networks may behave like space-filling curves, which further explains the strong expressivity they exhibit in practice. Although there is no direct evidence in support of this claim, it does shed light on a fundamental trade-off between the approximation capacity and the regularity of a mapping, especially when it is from a lower-dimensional space to a higher-dimensional space.

To make it clear, consider a mapping $f : \mathbb{R}^N \to L^2(D)$. Note that $\mathbb{R}^N$ and $L^2(D)$ have the same cardinality. According to the Schröder-Bernstein theorem (Folland, 1999), there exists a bijection $g$ between these two spaces. Therefore, if we do not require $f$ to be continuous, the lowest infimum that can achieve is always 0 since we can just simply take $f = g$. However, such a representation is definitely not what we want, as no differentiable space-filling curve can exist. On the other hand, if we assume that $f$ is globally Lipschitz continuous with some constant $K > 0$, the Hausdorff dimension of its image set $Im(f)$ is at most $N$ (Falconer, 1990).

In the main text, we have demonstrated that neural networks provide excellent approximations for $L^2(D)$ functions. However, this comes at the cost of compromising the attainability of the global

minimum (or even its existence). Although it is still unclear whether neural networks are already the optimal solution for this trade-off, we believe that delving deeper into this problem will advance the understanding of deep learning.

# D    PROOFS OMITTED IN SECTION 5

## D.1    PROOF OF THEOREM 4.1

*Proof.* Since $g : \mathbb{R}^N \to L^2(D)$ is radially unbounded, there exists $R > 0$ such that $\|g(\theta)\|_2 \geq 2$ for all $|\theta| \geq R$. Thus, we have

$$g^{-1}(\overline{Im(g)} \cap \mathcal{B}) \subset B_2$$

where $B_2 = \{\theta \in \mathbb{R}^N : |\theta| \leq 2\}$ and hence bounded. Using the fact that $g$ is continuous yields $g^{-1}(\overline{Im(g)} \cap \mathcal{B})$ is a closed, combining with its boundedness gives that $g^{-1}(\overline{Im(g)} \cap \mathcal{B})$ is compact, so its continuous image $\overline{Im(g)} \cap \mathcal{B}$. $\qquad\square$

## D.2    PROOF OF THEOREM 4.2

*Proof.* Without loss of generality, we assume $D = [0, 1]$. Consider the function sequence $\{\phi_n\}_{n=1}^{\infty} \subset L^1(D)$ where $\phi_n$ is given by

$$\phi_n = \left\{ \begin{array}{ll} 2^n, & x \in (1 - 2^{1-n}, 1 - 2^{-n}); \\ 0, & \textit{elsewhere}. \end{array} \right.$$

It has $\|\phi_m - \phi_n\|_1 = 2$ for all $m \neq n$, as $(1 - 2^{1-n}, 1 - 2^{-n}) \cap (1 - 2^{1-m}, 1 - 2^{-m}) = \varnothing$ when $m \neq n$.

Let $B_n \subset L^1(D)$ denote the ball centered at $\phi_n$ with radius $1 - \epsilon$. For any $\epsilon \in (0, 1)$, we have $B_n \cap B_m = \varnothing$ for any $m \neq n$. Furthermore, for any two functions $g_n \in B_n$ and $g_m \in B_m$, the distance between them is at least $2\epsilon$.

Suppose that for each $n \in \mathbb{N}$, there exists $h_n \in S$ such that $\|\phi_n - h_n\|_1 < 1 - \epsilon$, then $h_n \in B_n$ for all $n$. However, since $\|h_n - h_m\|_1 \geq 2\epsilon$ for all $m \neq n$, $\{h_n\}$ has no convergent subsequence. Thus, $S$ cannot be sequentially compact, which contradicts the fact that compactness and sequential compactness are equivalent in metric spaces.

As a consequence, there exists $n \in \mathbb{N}$ such that $\|\phi_n - h\|_1$ for all $h \in S$ and we complete the proof. $\qquad\square$

## D.3    PROOF OF LEMMA 4.1

*Proof.* It suffices to show that $S_1(1) \cap \mathcal{B}$ is non-compact. Consider the function sequence $\{\phi_n\}_{n=1}^{\infty} \subset \mathcal{B}$ where $\phi_n$ is given by

$$\phi_n = \left\{ \begin{array}{ll} 2^{2n}, & x \in (1 - 2^{1-n}, 1 - 2^{-n}); \\ 0, & \textit{elsewhere}. \end{array} \right.$$

Then we have $\|\phi_m - \phi_n\|_2 = \sqrt{2}$. According to the proof in the previous subsection, the set $\{\phi_n\}_{n=1}^{\infty}$ is not compact. Since $\{\phi_n\}_{n=1}^{\infty} \subset S_1(1) \cap \mathcal{B}$, $S_1(1) \cap \mathcal{B}$ is non-compact. $\qquad\square$

# E    ADDITIONAL RESULTS ON BURGERS' EQUATION

Figure 9 visualizes the predicted solution against the numerical solution at different $t$.

# F    SOME OPEN PROBLEMS

Here we list some open problems from our perspective:

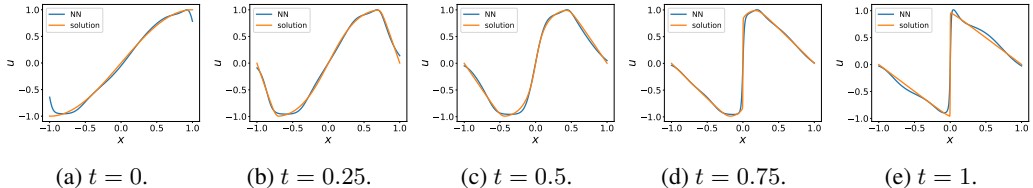

(a) $t = 0$.      (b) $t = 0.25$.      (c) $t = 0.5$.      (d) $t = 0.75$.      (e) $t = 1$.

Figure 9: Comparison of predicted and analytical solution of Burgers' equation at different time.

**Approximation theory**

- Does there exist a continuous mapping $f : \mathbb{R}^N \to L^1(D)$ for which we can find some $0 < M < 1$ such that for any $g \in \mathcal{B}$, it has $\inf_{h \in Im(f) \cap \mathcal{B}} \|g - h\|_1 \leq M$?

- If such mapping exists, what conditions does it have to satisfy?

- If such mapping does not exist, how far the neural network is from being the optimal approximator?

- Is there any connection between the amount of local minima and the expressiveness? For instance, given the number of parameters, can we say that the more expressiveness it possesses, the more local minima it must have?

**Classification**

- Given a data set, is there any better way to initialize the weights and biases more efficiently?

- Furthermore, given a target function, can we initialize the neural network more efficiently using the knowledge of its level sets?

- Can we also study the neural network training by looking into how the decision boundaries evolve?

- Can we determine the minimum width and depth of a network required to approximate a function based on the complexity of its level sets?

**Practical issues**

- Can we find a better activation function that does not suffer from the vanishing/exploding gradient issue? If possible, what requirements does it have to satisfy?

- Given a target function and a neural network, can we figure out whether the global minimum exists or not by running just a few iterations?

- Is it possible to extend the result of Theorem 4.1 to a more general case?

- Can we use an adaptive sampling technique similar to the adaptive mesh refinement in FEM to improve the performance of neural network approximation, especially in the case of discontinuous target functions?

- Usually, adaptive mesh refinement requires the flexibility of an approximator that allows for updates within a small area without affecting other regions. Typically, piece-wise linear/quadratic basis functions can handle this issue, but how can we determine if neural networks with complex structures can achieve the same capability?

