# OpenReview forum: "Understanding the Approximation Gap of Neural Networks"
_ICLR.cc/2024/Conference — ICLR 2024 Conference Withdrawn Submission_

### Official Review · Reviewer_Bouw · 2023-10-26

**Soundness:** 2 fair
**Presentation:** 2 fair
**Contribution:** 3 good
**Rating:** 6
**Confidence:** 3

**Summary:**

This paper studies the approximation gap of neural networks (NN) in scientific computing. The attempt is theoretical and the authors provided some novel perspectives, by studying the topology of the image set of NN and considering the practical assumption when the representation of floats has finite precision. Additionally, the authors found that increasing depth does not necessarily improve the accuracy in NN approximation. The paper also provides some numerical experiments to validate these claims.

**Strengths:**

1. The question is interesting: The deep learning (DL) methods have certain limitations compared with traditional numerical methods in solving PDE. Investigating how this limitation arises is an interesting research question and this work is properly motivated.

2. The findings and perspectives are novel: The authors prove several interesting and novel results, for example, increasing the depth does not necessarily improve the capability of NN on isolable sets. Studying the approximation of NN with the assumption that the float resolution is finite is realistic and novel.

3. Mathematical rigor: The theoretical claims in the paper are mathematically rigorous.

**Weaknesses:**

1. Nonconstructive result: The implication of the topology on the image set of NN is nonconstructive and hard to grasp. It is hard to understand how this relates to the NN approximation gap. A realistic example can help illustrate this point.

2. Presentation can be improved: It appears that the authors provide several independent causes for the NN approximation gap; and in the numerical experiments, it does not seem all the causes are verified. The authors can draw the connection more explicitly.

3. Limited impact: From the deep learning perspective, applications of NN in scientific computing are not mainstream. For example, the approximation gap is more likely when the PDEs only have discontinuous solutions as the authors claim. However, in applications of vision and natural languages, it is unsure whether such a discontinuity assumption holds. On the other hand, the result that approximating a discontinuous function using continuous NN is hard is not surprising.

Minor:
1. Section 3, paragraph 2: Let us $J(\theta)$ --> Let $J(\theta)$

2. The index for union is $\cup_{K=1}^\infty$ instead of  $\cup_{i=1}^\infty$?

3. A more standard notation for the ReLU function is ReLU instead of Relu.

4. L is used in Lebesgue measurable functions and neural network width.

**Questions:**

1. Are the results independent? For example, if an NN method fails to solve the PDE while traditional methods are able to handle it, could you attribute why the NN method fails as claimed in the paper?

2. While the result on the NN depth is interesting and novel, is it directly related to the approximation gap of NN in solving PDEs?

3. Suppose the assumptions do not hold in the paper: the PDEs have continuous solutions and the global minimum is not at infinity, will NN methods work for them?

Overall, more evidence supporting those points can strengthen this paper if understanding the approximation gap is the subject.

---

> ### Author Response · Authors · 2023-11-17
>
> Thanks for your feedback! Please see the following responses:
>
> Weaknesses:
>
> 1. A direct implication of non-compactness is that the global minimum may not exist, which then poses challenge to obtaining a good approximation numerically. The approximation gap is, even if a neural network is capable of approximating a function very well, we still have no way to realize a good representation via any usable techniques. The sigmoid neuron example in the Introduction is a simple illustration of this.
>
> 2. The results in our paper are not isolated; instead, they are correlated in a way that forms a chain leading to the approximation gap of neural networks. In Section 3, we provided a different perspective on the universal approximation theorem, emphasizing that the approximation power of neural networks is closely related to the non-compactness of their image sets, as demonstrated in Section 4. As a consequence, the global minimum may not exist due to such non-compactness. However, someone might argue that even if the global minimum does not exist, we can still achieve arbitrarily small error by employing very large weights. In Section 5, we demonstrate that it is impossible due to the machine precision and exponential decay of gradient. We also demonstrated that this issue cannot be simply resolved by increasing the depth alone, and we provide numerical examples to support this assertion.
>
> 3. We agree that the application of neural nets in scientific computing is not as popular as in CV and NLP, though there have been numerous works on it. In fact, we are skeptical about whether neural networks can match the performance achieved by classical methods in traditional fields. As you suggested, approximating a discontinuous function using continuous NN is hard is not surprising and we theoretically provide a lower bound for the attainable error in this paper. However, classical methods have already proved their strength in handling those problems, and the question we want to address is whether people continue pursuing this direction, given the fundamental limitations that hinder neural networks from working as well as traditional methods.
>
> And the questions:
>
> 1. Actually, there are numerous potential factors that can lead to a failure, such as incorrect loss function, poor initialization, wrong step size in gradient descent, etc.. Learning-based methods sometimes more or less yield a solution to the PDE, while at other times, they fail and generate garbage. Therefore, it is very difficult to locate the cause directly. Instead, our claim is that even if they provided a relatively reasonable solution, the absolute error between their solution and the true solution is still above certain threshold as demonstrated in our paper. Therefore, our result can be understood in the way that there is a gap for neural networks in solving PDEs, resulting in significantly higher errors compared to classical methods.
>
> 2. They are related in the way that for many PDEs in fluid mechanics, the level sets are highly disconnected and increasing the depth alone cannot resolve this issue.
>
> 3. The issue is that the global minimum is usually attained at infinity when the target function is discontinuous, leading to numerical difficulties. When the target function is sufficiently smooth, however, there is still no guarantee that the global minimum exists due to the non-compactness of image set. In the work "Characterizing possible failure modes in physics-informed neural networks" by Krishnapriyan et al., the absolute errors are still of order $\mathcal{O}(\sqrt{\Delta x})$ even if the solution is smooth.

---

> > ### Comment · Reviewer_Bouw · 2023-11-23
> > **Thanks for the response**
> >
> > I have read the response and appreciate it. The point that "The results in our paper are not isolated; instead, they are correlated in a way that forms a chain leading to the approximation gap of neural networks." is not well reflected in the paper; thus a revised writing highlighting this is desired.

---

### Official Review · Reviewer_WKkT · 2023-10-29

**Soundness:** 3 good
**Presentation:** 2 fair
**Contribution:** 2 fair
**Rating:** 5
**Confidence:** 2

**Summary:**

Summary. The authors study the approximation gap of neural networks. They derive that a bounded subset of the image set of a neural network does not necessarily have compact closure, especially in the case of discontinuous target functions. The authors also explain that the increasing depth alone is sometimes insufficient to improve the approximation accuracy.

**Strengths:**

Originality: The related works are adequately cited. One advantage of this paper is to study the approximation property of deep neural networks for some unusual functions and derive some related results. The main results in this paper will certainly help us have a better understanding of the universal approximation property of deep neural networks from a theoretical way. I have checked the technique parts and found that the proofs sound solid.

Quality: This paper is technically sound.

Clarity: This paper is not very clearly written. I find it is not easy to follow.

Significance: I think the results in this paper are not very significant, as explained below.

**Weaknesses:**

Some of the results in this paper are not quite interesting. For example, Theorem 3.1 derives a universal approximation theorem of two-layer sigmoid or Relu neural networks for $L^2 (D)$. Although it provides some results for the needed width, it is still not a significant contribution since the universal approximation of neural networks for $L^p$ was already known more than 30 years ago.

**Questions:**

1. Line 5 in Section 3,  Let us J(θ) denote -->  Let J(θ) denote.
2. Line 13 in Section 3,  $S_K(N))$ -->  $S_K(\mathbb{N})$.

---

> ### Author Response · Authors · 2023-11-17
>
> Thanks for your review. Regarding the specific question you raised on Theorem 3.1 (Universal Approximation Theorem), indeed it has been proposed decades ago as you suggested. However, the original derivation was rather in a formal manner using functional analysis/Fourier transform. In this paper, we derived it using step functions that are in the closure of the image set $Im(f)$, which further implies that the best approximation may not be achieved by any finite parameters. The main contribution of our work is demonstrating the non-compactness of the image set $Im(f)$ and elaborating how it is attributed to the practical approximation gap.
>
> Regarding $S_K(N)$, the symbol $N$ stands for the dimension of the parameter space $\mathbb{R}^N$, not the set of integers.

---

### Official Review · Reviewer_KVYb · 2023-10-30

**Soundness:** 2 fair
**Presentation:** 3 good
**Contribution:** 2 fair
**Rating:** 3
**Confidence:** 3

**Summary:**

This paper investigates the theory-practice gap employing neural networks to solve scientific computing problems: In contrast to the universal approximation theorems that ensures neural networks' good approximation ability (given sufficient size), neural networks suffer in situations that involve discontinuous or highly oscillatory solutions.

The paper shows the following:
1. For neural networks, the closure of the set of functions represented by neural networks is not compact (in the $L^2$ space), which implies that there is no global minimum in the loss landscape.

2. In such cases, finite precision in real computers prevents us from approaching the infimum at infinity, incurring some degree of inevitable $L^2$ loss.

3. The paper then turns to classification, studying the role of depth in isolating given data points (similar to the context of shattering). The paper shows that depth alone provides limited benefits in terms of the network's capability to isolate data points: if all shallow networks fail to isolate certain points, a network with an additional layer must also fail.

**Strengths:**

1. This paper studies the theory-practice gap present in scientific computing (e.g., solving PDEs using neural networks) in a mathematically rigorous way, and provides new insights.

2. The paper reads quite well, although there are some minor typos to be corrected:
- Below Eq (1), $\bigcup_{i=1}^\infty$ -> $\bigcup_{K=1}^\infty$?
- Proof of Theorem 3.1: in the ReLU, $y$ should be corrected to $\sigma(nx+1/2)-\sigma(nx-1/2)-\sigma(-nx+n-1/2)+\sigma(-nx+n+1/2)$.
- Image set $Im(f)$, effective region $\mathcal E(w,b)$ used without proper definition.
- Definition 5.2: shattered by $H$ -> $\mathcal S$?

**Weaknesses:**

The paper claims to investigate the approximation gap, but I am not really convinced why the main results should be surprising or whether the results are really relevant to approximation.

1. The paper claims that the bounded subset of the image set is compact for classical methods and non-compact for neural networks, and uses this fact to claim that neural networks have limited capability for approximating discontinuous functions. However, can compactness alone guarantee that a scheme is able to approximate discontinuity? I may not be very familiar with the classical approximation schemes mentioned here, but I know the Fourier series suffers from the Gibbs phenomenon, so I am doubtful if classical methods really deal with discontinuity well. Could you comment on this?

2. The intuition for non-compactness of neural network image set is roughly the following: sigmoid $x \mapsto \frac{1}{1+\exp(-\theta x)}$ cannot approximate the step function $1\\\{x \ge 0\\\}$ unless $\theta \to \infty$. But I wonder why this observation should be surprising?

3. I also do not see why we should be surprised about the finite precision result. With finite precision, we expect to see numerical errors anyway. Also, in contrast to the claim of $O(\frac{1}{\sqrt{p}})$ error in the introduction, Eq (2) seems to suggest that the $L^2$ error can be made small by making the grid size $\Delta x$ finer.

4. I am not 100% convinced of the connection of the classification result to function approximation. Section 5 is about isolability and shattering, which arise in the context of statistical learning theory. Being able to isolate each data point (separation) and assign **arbitrary** labels (gluing) seems to be a much stronger requirement than being able to approximate a given highly-fluctuating function. Hence, drawing conclusions on approximation from these results could potentially be misleading. For example, in the introduction, it is claimed that "This result suggests that the target function whose level sets are highly disconnected are naturally hard to approximate and increasing the depth alone is not sufficient to fit them well." However, this sounds like a contradiction to the well-known result in depth separation "Representation Benefits of Deep Feedforward Networks" by Telgarsky (2015), which shows that deep and narrow networks are much better at representing a highly-oscillating train of triangle waves compared to shallow and wide networks.

5. I am not convinced if the experiments corroborate and solidify the theoretical findings in the paper. Fitting complex functions using neural networks is a highly nonconvex optimization problem, and even if there is a perfect approximating solution, it is possible that optimization algorithms never reach it due to complications arising from nonconvexity. It seems difficult to claim that the poor performance of neural networks can be solely attributed to the limited approximation power.

**Questions:**

Please see the weaknesses section.

---

> ### Author Response · Authors · 2023-11-17
>
> Thanks for your detailed comments. Regarding your questions, please see the following items:
>
> 1. It is a good question. Aside from the Gibbs phenomenon, there is also a famous example called "Runge's phenomenon" in polynomial interpolation so we agree that some classical methods may have trouble as well when approximating discontinuous functions. However, for traditional schemes, those issues can be easily handled by using different set of basis functions. In particular, the Gibbs phenomenon can be eliminated by replacing Fourier series with Fejer kernel, and the Runge's phenomenon can be ameliorated by switching to Bernstein polynomials. However, there is no such measure for neural networks, as all of them suffer from the non-existence of global minima as we demonstrated, regardless of their size and activation function.
>
> 2. Actually, the example we showed in Introduction is just to help the audience to quickly grasp the idea. As suggested by reviewer oTBq, we could begin with a non-trivial example of parabolic PDE whose solution can be well-approximated by spectral methods because of their exponential convergence property in the function space. In contrast, achieving the same accuracy using neural networks is not possible due to the non-existence of global minima. However, we doubt if it is really fine for a paper to begin with such complicated discussions.
>
> 3. It can be understood in two ways: first, it should note that though machine precision is everywhere, it becomes a serious matter when the target function is not smooth. In fact, the error bound is derived when the norm of the gradient decays exponentially as weights approach infinity since the global minimum does not exist. Therefore, if the target function is sufficiently smooth so that there is a global minimum, weights do not need to approach infinity to attain it, thereby mitigating the effect of machine precision. Second, making $\Delta x$ finer directly incurs the curse of dimensionality, as the number of sample nodes grows exponentially with the dimension. One main claimed advantage of using neural networks over traditional methods is their capability of approximating functions with fewer data points, and our results provide a negative answer to it.
>
>     In particular, in Section 6, we solved the PDE using spectral method and neural network on the same grid, and result in different accuracy. This suggests that neural networks are more sensitive to the discretization error than classical methods.
>
> 4. Similar results were reported in works that are cited in our paper, such as Liang (2016) and Elbrächter (2019). In fact, this question is close to the next question on the approximation power of neural nets. In all those papers including the one you suggested, their claims are addressed through a carefully-designed example in which we happen to know the global minimum of approximation. However, having a low infimum does not justify the approximation power, as there is no concrete algorithm to reach it. Our training results on the Cantor set and Ackley function serve as such evidence.
>
> 5. In this paper, we aim to show that the approximation power involves two aspects: the capability to represent a target function and the difficulty of realizing that representation. Although the Universal Approximation Theorem states that neural networks satisfy the first part, and we showed that it is sometimes impossible to realize a good approximation due to the non-compactness of image set. In contrast, classical methods do not face the challenge of difficult realization, as they often have closed-form expressions for the best approximation (see Appendix B), and hence there is no need to find the global minimum through gradient-based algorithms. The situation that optimization algorithms never reach it due to complications arising from non-convexity itself, as you suggested, is itself a limitation of neural network approximation.
>
>     Just one more thing we want to mention, it is totally acceptable that neural networks are unable to converge towards global minima in applications of CV or NLP where on classical methods can handle. However, we should be very cautious when applying neural networks to scientific computing where classical methods have already exhibited and proved their full strength. The question we try to address is whether people should continue to use neural networks in a field where they are plagued by various issues that seldom affect traditional methods.

---

> > ### Comment · Reviewer_KVYb · 2023-12-01
> >
> > I appreciate the authors for their response. Some of my concerns were addressed, but some remain unsolved; allow me to elaborate more.
> >
> > 1, 2, 3. The authors mention that classical methods can also suffer problems from discontinuity, but there are fixes to it; in contrast, neural networks do not have such measures. However, what stops us from employing some additional components or tricks to fix this issue? It seems like there is already a line of research for handling such issues; when I google "discontinuity neural network", papers like "A Discontinuity Capturing Shallow Neural Network for Elliptic Interface Problems" (arxiv: 2106.05587) show up, and the networks presented therein may not suffer the problems such as the absence of global minima. So, it is still a little confusing to me why continuous neural networks being unable to approximate discontinuity is a serious and fundamental limitation.
> >
> > 4. As the authors mention, the papers Liang et al. 2016 and Elbrachter et al. 2019 are examples of papers demonstrating the "depth separation: deep and narrow networks have better approximation power than shallow and wide networks. Even with the authors' response, it is still hard for me to see why the discussion of the stronger task of isolability and shattering bears any decisive conclusion on approximation. The claim on hardness of approximation with deep networks should be better contextualized relative to the existing results on depth separation.
> >
> > 5. Although I agree with your claim that we have to be careful when applying NNs to scientific computing, I think the claim on two aspects of approximation power should be revised more carefully. The first aspect (around the universal approximation theorem) is what is commonly viewed as "approximation", and the second aspect (related to "finding" or "realizing" such good approximation) belongs to the category of "optimization", not "approximation" in my opinion. Hence, I believe that the poor experimental results should not be solely attributed to "approximation gap" discussed in the earlier sections of this paper, since this could be confusing or even misleading.
> >
> > In summary, although the authors' response clarified some parts of my concerns, there are some lingering issues which make me hesitate to raise my score at this time.

---

### Official Review · Reviewer_oTBq · 2023-11-03

**Soundness:** 2 fair
**Presentation:** 1 poor
**Contribution:** 2 fair
**Rating:** 3
**Confidence:** 3

**Summary:**

The paper is motivated by the empirical observation that neural networks are not well-suited to achieve high-performance for tasks in scientific computing and goes forth to propose reasons why this might be the case. The stated reasons and exploration in the paper have to do with the topology of the neural networks and also the discontinuity of sought-after solutions (e.g., in PDEs)

In summary, the paper takes a more realistic approach to the universal approximation theory that has been developed for neural networks, and study the so-called approximation gap that is empirically observed. The authors establish a connection between approximation and classification of level sets and show that to separate differently-labeled input points is handled by the width of the network while assigning same-labeled input points to the same region in space is handledg by the depth. However, if the target function has certain discontinuity properties, then the neural network's depth will not be enough to guarantee good accuracy results.

**Strengths:**

+well-motivated problem of understanding the approximation gap in practice and with theoretical justification

**Weaknesses:**

-the paper is written in a very confusing way, in that there is no clear connection between the technical aspect of what is proven, and the realistic interpretation of what the result actually implies for the practice of neural networks or their properties in the real-world. Simply saying that some set is not compact is not good enough.

-I find the weight-precision problem mentioned as a third contribution a bit misleading. Isn't it always a problem that we have finite machine precision?

-overall the paper has a nice collection of interesting results, but none of them is especially impactful. As such, I believe the paper does some incremental progress but doesn't have a "claim-to-fame" result that would be on par with an ICLR paper.

**Questions:**

Q: I believe for the presentation it would be helpful to start with a very simple example that showcases the difficulty of a neural network to solve a task (e.g. for a PDE). Then it would be nice to showcase how your results are informative in that simple example and actually give interpretation of the results. Currently, the presentation is very complicated.

---

> ### Author Response · Authors · 2023-11-17
>
> Thanks for the review. Please see the following responses to your concerns:
>
> 1. The connection between theoretical results and their practical implications is that, given the non-compactness of the image set, no good approximation may be achieved due to the non-existence of global minima. Consider the approximation problem $\min_{f \in S} \| f - g \|_2$ for some $g \in L^2(D)$ and $S \subset L^2(D)$. It is clear that there always exists $f^*$ when $S$ is compact. Indeed, such a best approximation $f^*$ may not exist when $S$ is not compact. All neural network training algorithms are gradient-based, which are expected to converge towards the global minimum. However, the global minimum itself may not even exist due to that non-compactness, then how much can we trust the results they yield?
>
> 2. In principle, finite machine precision can occur anywhere as we use machines, though it becomes more significant in certain cases than in others. In this paper, the error bound is derived when the norm of the gradient decays exponentially as weights approach infinity. In other words, if the global minimum exists, the weights do not necessarily need to approach infinity to attain it, thereby mitigating the effect of machine precision.
>
> 3. In fact, the results in our paper collaborate to form a fundamental claim regarding the approximation gap of neural networks: first, we show that the universal approximation power is somewhat due to the non-compactness of its non-compact set; second, we take a topological perspective to show that the compactness is indeed a fundamental distinction between neural networks and classical methods; third, one important implication of the non-compactness is the non-existence of global minima in certain cases, especially when the target function is not smooth enough; fourth, the approximation error is directly affected by the machine precision when global min does not exist; finally, we also showed that this issue cannot be resolved by simply increasing the depth alone.
>
> 4. Regarding the presentation, although numerous works attempting to solve PDEs using neural networks have been proposed, it would still be too difficult for those who are not familiar with the topic. We believe that providing an example with a single sigmoid neuron can help a larger audience quickly grasp the idea.